# Impulsivity as a Risk Factor for Addictive Disorder Severity during the COVID-19 Lockdown: Results from a Mixed Quantitative and Qualitative Study

**DOI:** 10.3390/ijerph20010705

**Published:** 2022-12-30

**Authors:** Maxime Pautrat, Antoine Le Guen, Servane Barrault, Aurélien Ribadier, Nicolas Ballon, Jean-Pierre Lebeau, Paul Brunault

**Affiliations:** 1EA7505 Education Ethique Santé, University of Tours, 37000 Tours, France; 2Department of General Practice, Tours Regional University Hospital, 37000 Tours, France; 3Qualipsy, EE 1901, Université de Tours, 37041 Tours, France; 4Laboratoire de Psychopathologie et Processus de Santé, Université de Paris, 92100 Boulogne Billancourt, France; 5CHRU (Centre Hospitalier Régional Universitaire) de Tours, Service d’Addictologie Universitaire, CSAPA-37, 37000 Tours, France; 6UMR 1253, iBrain, Université de Tours, INSERM, 37000 Tours, France; 7CHRU de Tours, Service d’Addictologie Universitaire, Équipe de Liaison et de Soins en Addictologie, 37000 Tours, France

**Keywords:** qualitative study, substance-related disorders, alcohol use disorders, behavioral addictions, addictive disorders, impulsivity, emotion regulation, COVID-19

## Abstract

Interindividual differences in personality traits, especially impulsivity traits, are robust risk factors for addictive disorders. However, their impact on addictive disorders during the COVID-19 lockdown remains unknown. This study assessed patients being followed for addictive disorders before the lockdown. We aimed to determine whether impulsivity traits (i.e., negative- and positive urgency) were associated with addictive disorders severity during the lockdowns. We also explored the patients’ subjective experiences, focusing on high versus low impulsivity. The quantitative study assessed 44 outpatients consulting for addictive disorders, for impulsivity, emotion regulation, anxiety/depression, and their addictive disorder characteristics, using self-administered questionnaires. In the qualitative study, six patients from the quantitative study were assessed using guided interviews. We observed that higher negative and positive urgencies were associated with addictive disorder severity. The subjective experiences of patients during the lockdowns differed according to their emotion-related impulsivity: high versus low. Low impulsive patients used online technologies more effectively to maintain follow-up, with more positive reappraisal. In contrast, highly impulsive patients reverted more frequently to self-medication with substances and/or behaviors, more social isolation, and found coping with negative emotions more challenging. Overall, the patient’s ability to cope with stressful events, like the COVID-19 lockdown, depended on their emotion-related impulsivity.

## 1. Introduction

Addictive disorders are diseases characterized by the interplay between three factors: the use of a specific substance or behavior that produces pleasure and/or reduces distress, individuals with vulnerability risk factors, and specific socioenvironmental context where the substance or the behavior is highly prevalent [1,2].

Since March 2020, the socio-environmental context in many countries has been brutally modified, following the decision by governments to enforce a lockdown due to the COVID-19 pandemic. This lockdown has severely constrained people’s daily lives, particularly those vulnerable to addictive disorders. These people lacked the time to adapt to these restrictions [3]. At the same time, the availability of addictive substances and behaviors remained largely unchanged [4]. Tobacco was available as usual. However, concerning alcohol, due to the closure of pubs, the sale of alcohol in shops was permitted in several European countries. For cannabis, due to the abruptness of the COVID-19 lockdown, the buying of stocks was limited [5]. However, European consumers continued to buy cannabis from outside of Europe without an observed increase in self-cultivation. In the United States, the availability of cannabis was unchanged with therapeutic tetrahydrocannabidiol considered “essential” and therefore available as usual. Concerning illicit drugs, the suspension of air traffic limited imports, as confirmed by a decrease in drug seizures by customs [3]. Despite this, stock management—by increasing prices and decreasing quality/purity—and the reorganization of sales (through social networks, home deliveries, etc.) meant that drug availability was not significantly changed. In Germany, notwithstanding the pandemic-imposed restrictions, and possible interpersonal differences, no change in buying and consumption of illicit drugs among polydrug users was observed [6]. What about addictive behaviors? Concerning gambling and gaming, a shift from sports betting to online poker, electronic sports viewing, and videogame streaming was observed [7]. The lockdown also induced a significant increase in internet use. Problematic internet use, such as online gaming and gambling and viewing of pornography, was of particular concern for mental health [8].

At the beginning of the first lockdown, a review summarized the psychopathological consequences of the socio-environment induced by the pandemic [9]. Boredom and isolation increased stress that promoted health risk behaviors, like substance use [10]. The switch from recreational to problematic use can occur during these social-isolated periods. Moreover, exposure to food through media publicity is known to contribute to food cravings and weight gain [11]. Patients with substance use disorders are more vulnerable during periods of confinement and associated stress, these conditions can exacerbate the disorder, if their substance is available [12,13]. However, not everyone coped the same during the lockdown. Understanding the potential interpersonal differences in addictive disorders vulnerability may provide a deeper understand of the mechanisms leading to varying relapse rates and would allow us to develop tailored treatments.

Interindividual differences in personality traits are robust risk factors for addictive disorders [14,15]. Among these personality traits, impulsivity, as defined by Evenden as, “actions without foresight that are poorly conceived, prematurely expressed, unnecessarily risky, and inappropriate to the situation” [16]. Impulsivity is an important risk factor for the transition from controlled to compulsive use of a substance/behavior and for relapse [17,18,19]. Furthermore, impulsivity is a major risk factor for both substance-related disorders and behavioral addictions [20,21,22,23,24]. An interesting comprehensive model of impulsivity that is useful for unraveling how various impulsivity traits contribute to addictive disorders was proposed by Whiteside & Lynam [25]. According to these authors, five distinct personality traits may be associated with impulsive behavior: negative urgency (i.e., the tendency to act rashly when faced with distress), positive urgency (i.e., the tendency to act rashly when faced with a very positive mood), lack of premeditation (i.e., the tendency to act without thinking), lack of perseverance (i.e., the inability to remain focused on a task), and sensation seeking (i.e., the tendency to seek out novel and thrilling experiences) [26]. Among these impulsivity traits, negative and positive urgency both relate to emotion dysregulation and to the tendency to respond reflexively to emotions. These impulsivity traits are associated with higher severity and poorer outcome in patients with substance use disorders [27,28]. Although quantitative studies have shown that impulsivity is a risk factor for addictive disorders [20,21,22,23,24], no study has investigated the subjective experiences of high-impulsive versus low-impulsive patients followed up for an addictive disorder during the COVID-19 lockdown. Among the impulsivity dimensions, we focused specifically on emotion-related impulsivity (i.e., negative urgency and positive urgency), as these dimensions are among the most important predictors for addictive disorders symptoms [29] and outcomes [28]. We hypothesized that these dimensions would be more relevant in the stress-related COVID-19 lockdown context. We did not focus on sensation seeking, because although sensation seeking is a known risk factor for substance use, its contribution in the maintenance of substance use, despite negative consequences, is not always observed [28].

We sought to address this issue by using a mixed quantitative and qualitative methodology, focusing on a broad range of addictive disorders (substance-related disorders and behavioral addictions), and focusing on emotion-related impulsivity (i.e., negative urgency and positive urgency). For the qualitative section, we divided patients with high versus low impulsivity, based on the approach proposed by Joël Billieux [30], that compared the corresponding negative urgency and positive urgency subscores of a given individual to the French norms. This approach allows the identification of patients with deviant scores for each dimension.

The aims of this mixed quantitative and qualitative study, conducted among patients that were followed up for an addictive disorder during the COVID-19 lockdowns, were twofold: (1) to determine whether some impulsivity traits (i.e., negative urgency and positive urgency) were associated with addictive disorders severity during the lockdowns (quantitative study), and (2) to explore the subjective experience of these patients during the lockdowns, by describing the experiences of patients with high versus low impulsivity (qualitative study).

## 2. Materials and Methods

### 2.1. Study Design

This cross-sectional study was designed with a mixed methodology: comprised of a quantitative and a qualitative study.

### 2.2. Participants

The quantitative study was conducted between July 2020 and December 2020 and the qualitative study between August 2021 and March 2022. Adults, living in France, with an addictive disorder that consulted in one of the following French outpatient centers were included: general practitioners (in the Centre-Val de Loire region), the Addiction Departments of the Georges Sand Hospital Center (Bourges) and the University Hospital of Tours, and the “Centre de Soins d’Accompagnement et de Prévention en Addictologie CICAT” (Chartres, France). All patients had a confirmed diagnosis by the Diagnostic and Statistical Manual of Mental Disorders (DSM)-5 criteria and were being followed up for at least one addictive disorder.

### 2.3. Quantitative Study

Quantitative data were collected using self-administered questionnaires. The questionnaire explored three main themes: patients’ socio-demographic characteristics, types and severity of addictive disorders, and symptoms associated with addictive disorders.

#### 2.3.1. Socio-Demographic Characteristics

Age, gender, marital status (married, in a relationship, or single), educational level (completion of at least a baccalaureate or a level below baccalaureate), and professional status were assessed using a self-administered questionnaire.

#### 2.3.2. Types and Severity of Addictive Disorders

The types of addictive disorders were assessed based on the medical record of their clinical assessments, using the DSM-5 criteria. We focused on the addictive disorders most likely to become dysbalanced during a lockdown (i.e., tobacco and alcohol use disorders, gambling disorder, and food addiction). We assessed the symptoms associated with these disorders as follows:For tobacco use disorders symptoms, we used the self-administered, validated French version of the Fagerström test for Nicotine Dependence (FTND) [31,32]. The FTND is composed of 6 items assessing nicotine and tobacco dependence and its severity. The total score ranges from 0 (not dependent) to 10 (highly dependent and severe).For alcohol use disorder, we used the Alcohol Use Disorder Identification Test (AUDIT) [33,34]. The AUDIT is a self-administered test that comprises 10 questions concerning the level of consumption, the associated symptoms of dependence, and the alcohol-related consequences. The AUDIT is scored from 0 to 40.For gambling disorders, we used the Canadian Problem Gambling Index (CPGI) [35]. This self-administered questionnaire includes 9 items, each score from 0 to 3. The overall score ranges from 0 to 27: 0 (no problem), 1–2 (low risk), 3–7 (moderate risk, and ≥ 8 problem gambling.For internet addictions, we used the French version of the Internet Addiction Test (IAT) [36,37]. This self-administered questionnaire includes 20 items scored from 0 to 100. The higher the score the more addicted the patient is to the internet.For food addictions, we used the modified Yale Food Addiction Scale 2.0 (mYFAS 2.0) [38,39]. This self-administered questionnaire comprises 13 items. The total mYFAS 2.0 score reflects the number of food addiction behaviors present: ranging from 0 to 11.

#### 2.3.3. Symptoms Associated with Addictive Disorders

Impulsivity, emotion dysregulation, and anxiety and depressive symptoms were assessed during the study.

Impulsivity was assessed using the French UPPS Impulsive Behavior Scale, short version (UPPS-P) [40]. The UPPS-P is a self-administered questionnaire based on the UPPS [25,41], with a measure of positive urgency [42]. The UPPS-P assesses 5 dimensions of impulsivity (namely, negative urgency, positive urgency, lack of premeditation, lack of perseverance, and sensation seeking). Each of the 20 items (4 items per dimension) are scored on a four-point Likert scale, ranging from 1 (strongly agree) to 4 (disagree strongly). The score for each dimension ranges from 0 to 16, with an overall score ranging from 0 to 64. The higher the score the higher the impulsivity.Emotion dysregulation was assessed using the brief version of the Difficulties in Emotion Regulation Scale (DERS-16) [43], comprised of 16 items and was adapted into French. The DERS-16 assesses various aspects of emotion regulation difficulties, including non-acceptance of negative emotions, inability to engage in goal-directed behaviors when distressed, difficulties controlling impulsive behaviors when distressed, limited access to emotion regulation strategies perceived as effective, and lack of emotional clarity. The DERS-16 total score ranges from 0 to 64. The score increases as the emotion dysregulation increases.Depressive and anxiety symptoms were evaluated using the French version of the Hospital Anxiety and Depression Scale (HADS) [44,45]. The HADS is a 14-item self-rating scale that assesses the severity of depression (7 items) and anxiety (7 items). Each item is scored between 0 (no, not at all) to 3 (yes, definitely). In this study, we used the total HAD depression subscores (ranging from 0 to 21) and the total HAD anxiety subscores (ranging from 0 to 21) as measures of depression and anxiety symptoms. The higher the score the higher the depression and anxiety.

### 2.4. Qualitative Study

For the qualitative study, patients were selected from those that completed questionnaires in the quantitative study. Purposive sampling was used to obtain patients with high and low impulsivity. Qualitative data were collected using guided interviews. The subjective experiences of patients were described according to the level of impulsivity: high versus low impulsivity. Patients with an impulsivity score higher than one standard deviation above the overall mean sub-score, for either negative urgency or positive urgency in the UPPS-P, were considered as having high impulsivity. The remaining patients were classified as having low impulsivity. Patients were recruited to maximize the variation in age, gender, and substance-related disorders versus behavioral addictions, and professional status. Participants were then divided into two groups: those with high impulsivity and those with low impulsivity. Interviews were conducted at the medical practices where patients were being followed up. The initial guided interview was developed by all the authors (see Appendix A). The guided interview was composed of an initial “icebreaker” question about how patients felt during the lockdown: providing them with the opportunity to talk about their experiences with respect to their addictive disorder(s), the role of caregivers, and their expectations of public authorities. Additional questions were added to explore concepts that emerged during the initial analysis. The interviews were recorded, transcribed, and anonymized.

### 2.5. Data Analysis

Quantitative analyses were conducted using SPSS version 22 (IBM Corp. Released 2013. IBM SPSS Statistics for Windows, Version 22.0 Armonk, NY, USA: IBM Corp.). All analyses were two-tailed; *p*-values < 0.05 were considered statistically significant. Quantitative data were described using descriptive statistics: percentages for ordinal variables and means with standard deviations for continuous variables. To determine the factors associated with either negative urgency or with positive urgency (UPPS-P subscores), we used Spearman’s correlation tests for independent continuous variables and Mann–Whitney non-parametric tests for independent ordinal variables.

The qualitative analysis was performed using two approaches: a phenomenological approach for some sections, including description of life experiences, and an approach inspired by grounded theory for sections focusing on social interactions. The verbatims were analyzed with respect to various categories. Those based on the phenomenological approach explored the patient’s feelings (including categories around emotions such as sadness or fear). While those based on social interactions explored their point of view on organizational aspects during the lockdown (including categories about family and professional organizations, their adaptations to care access, and their health constraints). The scientific validity criteria for grounded theory analysis were met. Indeed, 32 of 32 items in the Consolidated criteria for reporting qualitative research (COREQ) grid were completed. Our qualitative analysis included data triangulation and inductive analysis. We used the Sonal^®^ software for verbatim coding.

### 2.6. Ethical Considerations

This study was approved by the Ethics Committee of the University Hospital of Tours (IRB number: 2020 043). All procedures were performed in accordance with the ethical standards of the national and/or institutional research committee and with the 1964 Helsinki declaration and its later amendments or comparable ethical standards. All participants took part freely and voluntarily in the study. Informed consent was obtained from all individuals prior to study participation.

## 3. Results

### 3.1. Quantitative Study Results

#### 3.1.1. Socio-Demographic Characteristics, Types and Severity of Addictive Disorders, and the Associated Symptoms

Overall, we received 46 self-administered questionnaires. Among these, 44 were fully completed and included in the analysis. The data are summarized in Table 1. As for the main motives for treatment, most patients, 86.4% were followed up for at least one substance use disorder including 48.7% for alcohol, 29.9% for opioids, 18.2% for cocaine, and 11.4% for cannabis. In contrast, 22.7% were followed up for at least one behavioral addiction including 16% for food, 4.5% for gaming, and 2.3% for gambling. In terms of impulsivity (UPPS-P), the mean scores of certain subscores are noteworthy (maximum score of 4 for each subscore): 2.8 for negative urgency, 2.8 for positive urgency, and 2.6 for sensation seeking.

#### 3.1.2. Factors Associated with Negative and Positive Urgency

The correlations between impulsivity facets (i.e., negative and positive urgency), age, addictive disorder symptoms, emotion dysregulation, and anxiety and depression symptoms are presented in Table 2.

Higher negative urgency was associated with a higher score for tobacco use disorder, but not with alcohol use disorder, gambling disorder, internet addiction, and food addiction symptom scores. Negative urgency was also positively correlated with emotion dysregulation and anxiety symptoms. In contrast, negative urgency was not significantly associated with gender (Z = −0.55; *p* = 0.59), marital status (Z = −1.71; *p* = 0.09), being employed (Z = −0.76; *p* = 0.45), nor educational status (Z = −0.85; *p* = 0.40).

Higher positive urgency was associated with a higher food addiction score, but not with tobacco use disorder, alcohol use disorder, gambling disorder, nor internet addiction scores. Positive urgency was also positively correlated with emotion dysregulation and anxiety symptoms. However, positive urgency was not significantly associated with gender (Z = −0.80; *p* = 0.43), marital status (Z = −1.32; *p* = 0.19), being employed (Z = −0.74; *p* = 0.46), nor educational status (Z = −0.29; *p* = 0.77).

### 3.2. Qualitative Study Results

For the qualitative study, we interviewed 6 patients with a mean duration of the interview of 27 min. Among these, Patient #3 (P3) and Patient #5 (P5) represented those with high impulsivity. The other patients (Patient #1: P1, Patient #2: P2, Patient #4: P4, and Patient #6: P6) were classified as with low impulsivity. The patient characteristics are shown in Table 3.

#### 3.2.1. An Impression, in the Beginning, of Having Been Imprisoned at Home

After the lockdown was announced, patients with high impulsivity reported that they suffered due to the extreme limitations imposed on their daily lives, “*I cannot go and see my friends and family […], I cannot go and get cocaine as I want, finally I am really suffering living like this!*” (Patient #5). The reminiscence of a period of incarceration and the feeling of injustice made the lockdown unbearable, “*If I had not been imprisoned for 15 years, maybe I would not have felt this way, but now I find the lack of freedom even more insufferable, this really weighed me down*” (Patient #5). “*Even though I was not doing much, I knew that I had freedom, and they took this away from me*” (Patient #3).

The patients with a low impulsivity used the same terms to describe their limited, confined living space, “*Anyway, we were in prison! Even if we have nice cages with space, and everything we wanted, we were still confined at home*” (Patient #6). They also complained of a loss of freedom as if they had been condemned, “*We did not have chains, we did not have handcuffs, we did not have ankle chains, it was just that we could not go outside. It’s a violation of our basic freedom, which is nothing more than to be able to circulate, and this was very, very, very hard*” (Patient #6).

#### 3.2.2. A Shared Feel of Fear, but for Different Reasons

The fear “*to be without*” was especially expressed by patients with a low impulsivity, “*In the beginning, I thought the dealers would be restricted, but no. Also, even the product (heroin), I thought that after a certain time, it would not circulate as much, or they (the dealers) would not be able to obtain*” (Patient #4).

In contrast, patients with a high impulsivity expressed fear of the authorities, “*Since I was going walking with a bogus pretext, I was always afraid that I would come face to face with the police. If the cop was nice, it would be okay, the cop would say nothing, but if came face to face with a stupid cop. I was more scared of the police during the lockdown!*” (Patient #5).

#### 3.2.3. Common Core Values of Security

Several core values were common to patients with high and low impulsivity. Family support: “*My parents were there, they watched over me because they knew that I was in a bad state*” (Patient #3), “*I am lucky to have a partner that I get along very well with. It thanks to him that I held on*” (Patient #4). Those that were employed expressed the benefit of maintaining their work during the lockdown, “*Oh yes, thank goodness that I have a job, I do not know how people manage without a job, I would have lost my head*” (Patient #4). To be able to communicate with others, to ask them for help, and to support healthcare workers were mentioned in both the high and low impulsivity groups: “*Talking always does me good. With a psychologist, with a doctor, with someone from the centre, even with someone lambda, it does me good to talk*” (Patient #5), “*For example, there are things that I would tell my therapist that I do not tell my wife and so this loss of connection during the lockdown was difficult!*” (Patients #6).

#### 3.2.4. A Different Way of Consuming

All the patients interviewed maintained access to their psychoactive substance, or their behavior associated with their addiction, “*I thought dealers would not have the courage to go out, that I would see less of them than before the lockdown. Actually, nothing at all changed, they just met me wearing a mask*” (Patient #4). The concept of autotherapeutic consumption was shared: “*I think that I increased my consumption of heroin because the situation stressed me*” (Patient #1), “*I consumed to relax because I was very anxious and so this reduced the anxiety*” (Patient #3).

The patients with a low impulsivity asserted the desire to control their consumption but also indicated that the lockdown was a favorable period for weaning of their addiction, “*When this began (the lockdown), I told myself, maybe this will allow me to decrease my consumption*” (Patient #4).

In contrast, the patients with a high impulsivity mentioned a loss of control over their addiction: “*I went to see the doctor last night, because I do not have any more Subutex. I started shooting Subutex like a maniac*” (Patient #5), “*I was taking cortisone and I was being reckless with the benzodiazepines […] the pharmacist that served me saw that I was completely lost, and she asked me, what’s wrong? It is my mother who told me this*” (Patient #3).

#### 3.2.5. A Positive Experience, an Opportunity for a Better Life, Was More Evident in Low Impulsivity Patients

Patients with low impulsivity seemed to have had a more positive experience of the lockdown, than high impulsivity patients. They talked about the lockdown as if it were a holiday, “*The first month the weather was good, so this was a phase that I would consider as ‘cool’. I was with my family. I was at home, and I could eat whatever I wanted, it was the best*” (Patient #6). Daily life at home was not a source of conflict. “*It went well, we did not argue more than usual, and everything went well*” (Patient #4) There was a calm atmosphere. “*I can understand that some people lost their heads. Anyway, I was relatively calm*” (Patient #2). For certain patients the lockdown was an opportunity to improve their lives: “*I am not sure that I am going to continue (in the food business). There is a human, social side that I miss, I did this for more than 15 years, but (in the temporary work/factory) I have my weekends, I am paid more, I lose socially but I gain on the other side*” (Patient #4).

#### 3.2.6. Boredom, the Solitude, and the Weariness Was Expressed by High Impulsivity Patients

Loneliness and a profound boredom were conveyed by patients with high impulsivity: “*Sometimes I was really bored*” (Patient #5). “*I withdrew a little […] I listened to a lot of music; I watch lots of films. These were the main things that I did and other than that I did not do much*” (Patient #3). As the lockdown continued a state of weariness arose “*People always swept this under the rug, but I told them to stop, I spoke about other things, because I was tired of talking about the coronavirus*” (Patient #5). “*I was always concerned about the others, I was not going out, I was closed and completely paranoid*” (Patient #3). The lockdown seemed to be reinforced by an inability to adapt to the new technology: “*I don’t really know how to use the internet very well, I can’t seem to get into it, I am not from the internet generation, it is an unknown world for me. I regret this because I would like to be able to communicate by internet and do all this*” (Patient #5).

## 4. Discussion

This mixed qualitative and quantitative study found that among patients followed up for an addictive disorder during the COVID-19 lockdown, their subjective experiences differed according to their emotion-related impulsivity: high versus low. Although both groups reported the same stressful events during the lockdown (i.e., limitation in personal freedom, negative emotions like fear, and barriers to physical outpatient follow-up), they differed in the way they coped. Low impulsive patients made better use of online technologies to maintain follow-up and with more positive reappraisal. In contrast, highly impulsive patients reverted to self-medication with substances and/or behaviors, more social isolation, and found coping with negative emotions more challenging. In the quantitative study, we observed that higher negative and positive urgencies were associated with the severity of some addictive disorders.

The qualitative methodology allowed us to identify factors common among patients with high and low impulsivity, but also to distinguish the experiences, and protective and risk factors associated with high and low impulsivity. All the patients shared the sensation of a loss of freedom. Those that had been imprisoned considered the lockdown more as an injustice. The feeling of fear was also shared by the patients, but for different reasons. The less impulsive patients were more afraid that they would not have access to enough of their addictive substance. In contrast, those more impulsive, never doubted that they would have enough of their additive substance, even if they had to violate the imposed restrictions. They were more afraid of being controlled by the authorities. This was particularly true for a patient that had served 15 years in prison. A possible explanation is the social deafferentation hypothesis where complex changes in the brain can be induced by social withdrawal, prompting more impulsive behaviors [46]. The “dose” of social withdrawal, which is potentially psychotogenic, depends on the individual’s initial level of social involvement. This observation should be considered in patients with a history of social withdrawal in the event of further lockdowns.

Family, employment, and therapeutic relationship were protective elements during the lockdown, expressed by all patients. The patients without a pathological impulsivity score were also those that were employed and that expressed positive experiences during the lockdown. These results are consistent with a study, based on social media posts on “Reddit” (a forum in the United States), that discussed personal experiences of people who used drugs during the COVID-19 pandemic [47]. Though the pandemic negatively impacted existing coping strategies and access to formal support services, a minority of patients viewed lockdown and quarantine as an opportunity to lower or stop their substance use.

Patients with a high impulsivity reported that they lost control of their consumption and felt bored, lonely, and weary. This group of patients were more vulnerable [48]. They had fewer resources available to cope with stress, prior to the lockdown. Furthermore, during lockdown, their strategies to adjust to stress were more limited. They also reported that in response to being bored, isolated, and alone, they resorted to more frequent substance use to diminish their anxiety [10].

It is important to remember that social distancing does not necessarily mean social isolation. A study has described how social isolation causes psychological harm that hinders positive behavioral development [49]. Reducing “testing-social” opportunities could reduce “reality-monitoring” strategies and cause altered perceptions and behavioral disorders. In this study, conducted in the prison environment, the authors indicated how psychologists, policy makers, and others can contribute to creating a more effective and humane justice system. The same applies to the health system in the context of a pandemic lockdown.

During the lockdown, the patients wanted to maintain social contacts and a therapeutic relationship with their caregivers. However, patients with high impulsivity seemed to be less confident using new technology. Telehealth options, especially those that promote social connections are often presented as useful for patients with addiction [50,51]. But certain patients may have limited access to these digital technologies, preventing them from benefiting from online mental health services. Moreover, during the COVID-19 lockdown, a Chinese study recalled that the quality assurance of these online services remains problematic, and that the effectiveness of online mental health interventions, in low and middle-income countries, has not yet been rigorously evaluated [52]. These factors could participate in exacerbating mental health disparities.

### Strengths and Limits of the Study

Concerning the quantitative study, the length of the questionnaire could have limited the number of complete questionnaires obtained. The lack of data meant that less prevalent addictions, such as gambling, were not well represented in the study.

For the qualitative study, the use of a dual approach, comprising phenomenological and grounded theorization analysis, was justified by the search for the conceptualization of patients’ experiences [53]. The COREQ for grounded theorizing research were respected at each stage of the qualitative study [54]. However, within this methodology there is potential for interpretation and confirmation bias. The use of a strictly inductive analysis and the triangulation of data allowed us to limit the influence of the investigators’ subjectivity and preconceived ideas. The study was a multidisciplinary collaboration between psychiatrists, addictologists, general practitioners, and sociologists. This diversity of healthcare professionals enriched the data analysis.

The purposive sampling of the study, although limited, provided the necessary diversity and sufficient data to illustrate patients with high and low levels of impulsivity. This qualitative research is one of the first to focus on the patients’ experiences with substance use disorders during a pandemic lockdown. In the literature, data during social isolation periods have almost exclusively focused on imprisoned patients or those living in isolation, such as during explorations or those living in submarines [55,56,57,58]. Recent qualitative studies on the impact of the COVID-19 pandemic on the mental health of patients with substance use disorders included mostly harm reduction workers or healthcare workers, and not directly patients [59]. A very recent qualitative study interviewed patients and harm reduction workers, using a thematic analysis approach [60]. This made it possible to describe the daily life of the patients, including practical aspects, and highlighted the key infrastructure challenges inherent in addiction prevention and treatment continuum. Our study, with its phenomenological and theoretical approach, explored the patients’ emotions and their experiences, as well as the correlation between their level of impulsivity and addiction disorders. Finally, our results should not be extrapolated to different population from that under study, i.e., patients being followed up in a healthcare structure and who volunteered to participate in our study. Our results need to be validated in a larger more diverse population.

## 5. Conclusions

Overall, our results support the idea that impulsivity, and especially emotion-related impulsivity, may be a key variable for studying interpersonal differences in addictive disorder vulnerability. Even though patients with high or low emotion-related impulsivity shared the same stressful life events, and other subjective experiences, during the COVID-19 lockdown, they differed in the way they coped with these events. Considering interindividual differences in impulsivity may help to improve the tailoring of treatment for patients with addictive disorders during stressful life events. Further studies will be performed to evaluate the long-term psychological effects and evolution of addictive disorders after a lockdown period.

## Figures and Tables

**Table 1 ijerph-20-00705-t001:** Summary of data collected during the quantitative study.

	Data Collected	All Participants ^1^ (*n* = 44)
Socio-demographic characteristics		
	Age (years)	45.5 ± 11.0
	Sex (male)	21 (47.7%)
	Marital status (married or in a relationship)	17 (38.7%)
	Educational level (completion of at least a baccalaureate)	27 (61.4%)
	Professional status (currently employed)	20 (45.5%)
Current use of substance/behavior		
	Alcohol (yes)	30 (68.2%)
	Tobacco (yes)	27 (61.7%)
	Gambling (yes)	6 (13.6%)
	Internet (yes)	35 (79.5%)
Addictive disorders severity		
	Alcohol use disorder (AUDIT total score, range 0–40)	11.6 ± 13.1
	Tobacco use disorder (FTND total score, range 0–10)	4.6 ± 4.2
	Gambling disorder (CPGI total score, range 0–27)	1.0 ± 3.2
	Internet addiction (IAT total score, range 0–100)	27.3 ± 17.3
	Food addiction (mYFAS 2.0 total score, range 0–11)	1.5 ± 2.4
Emotion dysregulation (DERS total score, range 0–64)		6.4 ± 5.0
Impulsivity subscores (UPPS-P)		
	Negative urgency (score range: 1–4)	2.8 ± 0.7
	Positive urgency (score range: 1–4)	2.8 ± 0.6
	Lack of premeditation (score range: 1–4)	2.1 ± 0.7
	Lack of perseverance (score range: 1–4)	2.2 ± 0.6
	Sensation seeking (score range: 1–4)	2.6 ± 0.6
Anxiety and depression subscores (HAD)		
	Anxiety symptoms	9.1 ± 5.5
	Depression symptoms	7.2 ± 5.0

^1^ The data are presented as mean ± standard deviation or as number with percentage. AUDIT: Alcohol Use Disorder Inventory Test; CPGI: Canadian Pathological Gambling Inventory; DERS: Difficulties in Emotion Regulation Scale, 16 items; FTND: Fagerström Test for Nicotine Dependence; HAD: Hospital and Anxiety Depression Scale; IAT: Internet Addiction Test; mYFAS 2.0: modified Yale Food Addiction Scale, DSM-5 version; UPPS-S: UPPS Impulsive Behavior Scale, short version.

**Table 2 ijerph-20-00705-t002:** Correlations between negative and positive urgency, addictive disorder symptoms, emotion dysregulation, and anxiety and depression symptoms.

	1	2	3	4	5	6	7	8	9	10	11
1. Negative urgency (UPPS-P)	-										
2. Positive urgency (UPPS-P)	0.54 ***	-									
3. Age	−0.24	0.06	-								
4. Alcohol use disorder (AUDIT)	0.10	0.05	0.04	-							
5. Tobacco use disorder (FTND)	0.30 *	0.21	−0.11	0.04	-						
6. Gambling disorder (CPGI)	−0.14	−0.12	0.18	0.34 *	0.18	-					
7. Internet addiction (IAT)	−0.02	−0.03	−0.29	−0.21	−0.26	−0.04	-				
8. Food addiction (mYFAS 2.0)	0.28	0.35 *	−0.07	−0.04	−0.12	0.01	0.26	-			
9. Emotion dysregulation (DERS-16)	0.67 ***	0.37 *	−0.16	0.29	0.14	−0.19	0.04	0.24	-		
10. Anxiety symptoms (HAD)	0.40 **	0.33 *	−0.22	0.40 **	−0.01	0.08	0.06	0.33 *	0.49 ***	-	
11. Depression symptoms (HAD)	0.19	0.03	−0.12	0.32 *	−0.02	0.04	0.13	0.18	0.40 **	0.48 ***	-

Note. We used Spearman’s correlation tests because some variables did not meet normality assumptions. * *p* < 0.05; ** *p* < 0.01; *** *p* < 0.001. AUDIT: Alcohol Use Disorder Inventory Test; CPGI: Canadian Pathological Gambling Inventory; DERS: Difficulties in Emotion Regulation Scale, 16 items; FTND: Fagerström Test for Nicotine Dependence; HAD: Hospital and Anxiety Depression Scale; IAT: Internet Addiction Test; mYFAS 2.0: modified Yale Food Addiction Scale, DSM-5 version; UPPS-S: UPPS Impulsive Behavior Scale, short version.

**Table 3 ijerph-20-00705-t003:** The characteristics of patients included in the qualitative analysis.

Patient	Sociodemographic Characteristics	Main Addictive Disorder(s) ^1^	Other Addictive Disorder(s) ^2^	Impulsive (High or Low)	Negative Urgency Score ^3^	Positive Urgency Score ^3^	Depression Subscore ^3^
1	38 years old, female, single, and employed	Heroin use disorder	Tobacco use disorder	Low	2.75 (+0.06 SD)	2.50 (−0.39 SD)	13 (+1.19 SD)
2	36 years old, female, married, and unemployed	Heroin and cocaine use disorders	Tobacco use disorder	Low	2.50 (−0.29 SD)	2.75 (+0.05 SD)	0 (−1.43 SD)
3	41 years old, male, single, and unemployed	Opioid use disorders	Benzodiazepine use disorder	High	3.25 (+0.77 SD)	4.00 (+2.23 SD)	6 (−0.22 SD)
4	35 years old, female, married, and employed	Heroin use disorder	Tobacco use disorder	Low	2.50 (−0.29 SD)	2.75 (+0.05 SD)	6 (−0.22 SD)
5	56 years old, male, single, and unemployed	Cocaine use disorder	Tobacco use disorder	High	3.25 (+0.77 SD)	4.00 (+2.23 SD)	10 (+0.59 SD)
6	45 years old, male, married, employed	Food addiction	None	Low	2.50 (−0.29 SD)	3.00 (+0.48 SD)	8 (+0.18 SD)

^1^ Main addictive disorder(s) were the main motive for the consultations; ^2^ Other addictive disorder(s) were the other addictive disorders diagnosed during the initial consultation; ^3^ The values indicated are relative to the mean score and the standard deviation (SD) for the corresponding subscores obtained during the quantitative study: mean negative urgency in our full sample was 2.7 ± 0.7, mean positive urgency was 2.7 ± 0.6, and mean depression score was 7.1 ± 5.0.

## Data Availability

All data are available upon reasonable request from the corresponding author.

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
