# Peer review of "Impulsivity as a Risk Factor for Addictive Disorder Severity during the COVID-19 Lockdown: Results from a Mixed Quantitative and Qualitative Study"

_ijerph, 2022, doi:10.3390/ijerph20010705_

Round 1
Reviewer 1 Report
There are some minimal errors that should be checked.
In the abstract,
Only one comma is missing in line 27, and one point in line 31
In Materials and Methods section, in 2.2 Participants, it does not say, where are participants from, if they are French or another nationality, authors should write where participants are from.
Author Response
Itemized list of reviewers comments with corresponding revisions or responses
Manuscript ID: ijerph-2067518
Decision: Major revision
Reviewer#1 Comments
Comment#1. There are some minimal errors that should be checked. In the abstract, only one comma is missing in line 27, and one point in line 31
Reply: Thank you for your comment; we have updated the manuscript (Lines 27 and 31).
Comment#2. In Materials and Methods section, in 2.2 Participants, it does not say, where are participants from, if they are French or another nationality, authors should write where participants are from.
Reply: We do not have details concerning the nationalities of these patients, but they all live in France. We have updated the manuscript (Line 129).
“Adults, living in France, with an addictive disorder that consulted in one of the following French outpatient centers were included…”
Reviewer 2 Report
Despite the substantial negative impact the COVID-19 pandemic had on the world, it did provide a unique opportunity to study differences in individual responses to adapting to significant life-altering changes that, in many ways, were shared across people. It also allowed us to improve our understanding of protective and risk factors for adapting to the psychological fallout from these changes, especially among those deemed most vulnerable. So, studying how impulsivity impacted behaviors and symptom severity among those diagnosed with an addictive disorder is important, given its demonstrated relationship. Unfortunately, however, this study has many limitations that undermine enthusiasm for it, which are listed below.
1. The rationale for only considering positive and negative urgency needs to be clarified and further developed. Is there reason to believe other impulsivity traits would not relate to symptom severity within this context? Only including the other negative and positive impulsivity traits makes the paper seem incomplete, especially when sensation seeking is mentioned within the text and all traits are reported in Table 1. Even if the additional traits were to be added to the quantitative analysis, a strong rationale should then be included describing why patients were divided into high vs. low impulsivity for the qualitative section.
2. The qualitative section highlights the primary addictive disorder of the participant. With one exception, all were diagnosed with a substance use disorder other than tobacco and alcohol. The other was reprted to have problematic eating. Is there reason to believe that this is an unusual sub-sample? And as a result, is there a reason that their responses would (or would not) reflect how the majority of the sample? Likewise, if ~30% of the sample had OUD, would we expect an increase in severity for AUD or TUD?
Also, how are we to interpret the combination of findings between the lack of relationship between impulsivity and severity with what is reported in the qualitative results?
In addition, it needs to be clarified how these six participants were ultimately chosen. Please provide more information about this process.
3. The limitation section mentions that the length of the survey might have had an impact on the sample size. Please provide information regarding how many participants were dropped because of incomplete data.
4. There is no description of the Internet Addiction measure within the Methods section. Please provide.
5. In general, please provide more information describing each measure. At least one was modified. The possible range of scores would provide greater context in interpreting the means and SDs.
In general, a more robust rationale for the study's approach and design needs to be provided with a clear and well-supported description and discussion of the findings between the quantitative and qualitative and how they relate to each other.
Minor point(s): please review the manuscript to ensure consistency in formatting. For example, when quoting the participants, italics were used with the exception of section 3.1.3.
Author Response
Itemized list of reviewers comments with corresponding revisions or responses
Manuscript ID: ijerph-2067518
Decision: Major revision
Reviewer#2 Comments
Despite the substantial negative impact the COVID-19 pandemic had on the world, it did provide a unique opportunity to study differences in individual responses to adapting to significant life-altering changes that, in many ways, were shared across people. It also allowed us to improve our understanding of protective and risk factors for adapting to the psychological fallout from these changes, especially among those deemed most vulnerable. So, studying how impulsivity impacted behaviors and symptom severity among those diagnosed with an addictive disorder is important, given its demonstrated relationship. Unfortunately, however, this study has many limitations that undermine enthusiasm for it, which are listed below.
Comment#1. The rationale for only considering positive and negative urgency needs to be clarified and further developed. Is there reason to believe other impulsivity traits would not relate to symptom severity within this context? Only including the other negative and positive impulsivity traits makes the paper seem incomplete, especially when sensation seeking is mentioned within the text and all traits are reported in Table 1. Even if the additional traits were to be added to the quantitative analysis, a strong rationale should then be included describing why patients were divided into high vs. low impulsivity for the qualitative section.
Reply: We have modified the manuscript in response to this comment.
To clarify, we have updated the introduction as follows (Lines 100-107):
“Among the impulsivity dimensions, we focused specifically on emotion-related impulsivity (i.e., negative urgency and positive urgency) as these dimensions are among the most important predictors for addictive disorders symptoms [29] and outcomes [28]. We hypothesized that these dimensions would be more relevant in the stress-related COVID-19 lockdown context. We did not focus on sensation seeking, because although sensation seeking is a known risk factor for substance use, its contribution to maintain substance use despite negative consequence is not always observed [28].”
Furthermore, according to some authors, sensation seeking may be considered as a risk factor for substance use or an effect of some drug use rather than a risk factor for the transition between the use and compulsive use (Belin & Deroche-Gamonet, 2012; Ersche et al., 2010).”
In addition, we have updated the introduction (Lines 111-115) to clarify, as follows:
“For the qualitative section, we divided patients with high versus low impulsivity based on the approach proposed by Joël Billieux [30] that compared the corresponding negative urgency and positive urgency subscores of a given individual to the French norms. This approach allows the identification of patients with deviant scores for each dimension.”
Comment#2. The qualitative section highlights the primary addictive disorder of the participant. With one exception, all were diagnosed with a substance use disorder other than tobacco and alcohol.
Reply: Regarding the description of our sample in terms of the types of addictive disorders, we differentiated between the addictive disorders that were the main motives for consultations and other addictive disorders. In Table 3 we also added descriptive data regarding the other addictive disorders for each patient. In this way, we provide a clearer indication of the generalizable of our sample to the target population.
The other was reported to have problematic eating. Is there reason to believe that this is an unusual sub-sample?
Reply: From a quantitative point of view, you are correct: adding a single patient with such a specific addictive disorder could create an unusual sample. But from a qualitative point of view, it is relevant to present a sample with a wide variety of addictive disorders.
And as a result, is there a reason that their responses would (or would not) reflect how the majority of the sample? Likewise, if ~30% of the sample had OUD, would we expect an increase in severity for AUD or TUD?
Reply: We hope that the modifications made to Table 3 will better reflect our sample. Our qualitative study concerned 6 patients. This small sample size and the known heterogeneity of addictive disorders means that it is difficult to use our results to generalize. However, our study does provide information to better understand and to develop strategies for patients with addictive disorders during stressful situations. We also clearly indicate in lines 468-469 that, “Our results need to be validated in a larger more diverse population.”
Also, how are we to interpret the combination of findings between the lack of relationship between impulsivity and severity with what is reported in the qualitative results?
We agree with the reviewer, the relationship between impulsivity and severity was not the same using a quantitative and qualitative approach. This may be due to the lack of power of statistical power in the quantitative study where only 44 questionnaires were analyzed. Another reason may be the heterogeneity of the sample, with various substances use disorders, making it difficult to identify a relationship between impulsivity and severity. However, we feel this justifies the use of a mixed approach. Indeed, data from qualitative study differed from those obtained from quantitative study. These data were complementary and provided us with a broader understanding of these patients. Finally, the aim of our study was to characterize the impulsivity of patients with addictive disorders during the Covid lockdown, by a psychometrics and a qualitative approach, and not necessarily to establish a causal link between impulsivity and substance use disorder severity.
In addition, it needs to be clarified how these six participants were ultimately chosen. Please provide more information about this process.
Reply: Patients included in the qualitative data were selected by the authors from patients who had completed the quantitative questionnaires. The patients in the qualitative study were selected using a purposive sampling, as indicated in line 452, to optimize diversity. The following text has been added in the methods (Section 2.4 Qualitative study lines 196-198) to clarify the patient selection for the qualitative study:
“For the qualitative study, patients were selected from those that completed questionnaires in the quantitative study. Purposive sampling was used to obtain patients with high and low impulsivity.”
Comment#3. The limitation section mentions that the length of the survey might have had an impact on the sample size. Please provide information regarding how many participants were dropped because of incomplete data.
Reply: Among the questionnaires that we received, 2 questionnaires were incomplete and were not included in the quantitative study. Unfortunately, we lack data concerning the number of patients that obtained the questionnaire but did not complete them. The study questionnaires were self-administered questionnaires that were available in the waiting room. Patients were free to respond or not to the questionnaires.
The Section 3.1.1 line 246-247 has been modified as follows:
“Overall, we received The46 self-administered questionnaires. Among these, 44 waswere fully completed and included in the analysisby 44 patients.”
Comment#4. There is no description of the Internet Addiction measure within the Methods section. Please provide.
Reply: We thank the reviewer for this comments that will improve the manuscript’s quality. This was indeed lacking. We added a section to describe the Internet Addiction Test and more information regarding this measure in lines 163-165, as follows:
“For internet addictions, we used the French version of the Internet Addiction Test (IAT) [36,37]. This self-administered questionnaire includes 20 items and scored from 0 to 100. The higher the score the more addicted the patient is to the internet.”
Comment#5. In general, please provide more information describing each measure. At least one was modified. The possible range of scores would provide greater context in interpreting the means and SDs.
Reply: Thank you for your comment; we revised the text accordingly by providing more information for each measure, as follows:
Section 2.3.2, lines 146-168
“The types of addictive disorders were assessed based on medical record of their clinical assessments using the DSM-5 criteria. We focused on the addictive disorders most likely to become dysbalanced during a lockdown (i.e., tobacco and alcohol use disorders, gambling disorder, and food addiction). We assessed these disorders using the following self-administered questionnaires: the validated French version of the Fagerström test for Nicotine Dependence (FTND) [29,30] for tobacco use disorder, the Alcohol Use Disorder Identification Test (AUDIT) [31,32] for alcohol use disorder, the Canadian Problem Gam-bling Index (CPGI) for gambling disorder (Ferris & Wynne, 2001), and the modified Yale Food Addiction Scale 2.0 (YFAS 2.0) [33,34] for food addiction.We assessed the symptoms associated with these disorders as follows:
- For tobacco use disorders symptoms, we used the self-administered, validated French version of the Fagerström test for Nicotine Dependence (FTND) [31,32]. The FTND is composed of 6 items assessing nicotine and tobacco dependence and its severity. The total score ranges from 0 (not dependent) to 10 (highly dependent and severe).
- For alcohol use disorder, we used the Alcohol Use Disorder Identification Test (AU-DIT) [33,34]. The AUDIT is a self-administered test that comprises 10 questions con-cerning the level of consumption, the associated symptoms of dependence, and the alcohol-related consequences. The AUDIT is scored from 0 to 40.
- For gambling disorders, we used the Canadian Problem Gambling Index (CPGI) [35]. This self-administered questionnaire includes 9 items, each score from 0 to 3. The overall score ranges from 0 to 27: 0 (no problem), 1-2 (low risk), 3-7 (moderate risk, and ≥8 problem gambling.
- For internet addictions, we used the French version of the Internet Addiction Test (IAT) [36,37]. This self-administered questionnaire includes 20 items and scored from 0 to 100. The higher the score the more addicted the patient is to the internet.
- For food addictions, we used the modified Yale Food Addiction Scale 2.0 (mYFAS 2.0) [38,39]. This self-administered questionnaire comprises 13 items. The total mYFAS 2.0 score reflects the number of food addiction behaviors present: ranging from 0 to 11.”
Section 2.3.3, lines 172-194
“Impulsivity, emotion dysregulation, and anxiety and depressive symptoms were assessed during the study. Impulsivity was assessed using the French UPPS Impulsive Behavior Scale, short version (UPPS-P) [35]. The UPPS-P is a self-administered questionnaire based on the UPPS [25,36] and one measure of positive urgency [37], and is comprised of 20 items. We assessed emotion dysregulation using the brief version of the Difficulties in Emotion Regulation Scale (DERS-16) [38], comprised of 16 items and was adapted into French. Finally, depressive and anxiety symptoms were evaluated using the French version of the Hospital Anxiety and Depression Scale (HADS) [39,40].
- Impulsivity was assessed using the French UPPS Impulsive Behavior Scale, short version (UPPS-P) [40]. The UPPS-P is a self-administered questionnaire based on the UPPS [25,41] with a measure of positive urgency [42]. The UPPS-P, assesses 5 dimensions of impulsivity (namely, negative urgency, positive urgency, lack of premeditation, lack of perseverance, and sensation seeking). Each of the 20 items (4 items per dimension) are score on a four-point Likert scale, ranging from 1 (strongly agree) to 4 (disagree strongly). The score for each dimension ranges from 0 to 16, with an overall score ranging from 0 to 64. The higher the score the higher the impulsivity.
- Emotion dysregulation was assessed using the brief version of the Difficulties in Emotion Regulation Scale (DERS-16) [43], comprised of 16 items and was adapted into French. The DERS-16 assesses various aspects of emotion regulation difficulties, including non-acceptance of negative emotions, inability to engage in goal-directed behaviors when distressed, difficulties controlling impulsive behaviors when distressed, limited access to emotion regulation strategies perceived as effective, and lack of emotional clarity. The DERS-16 total score ranges from 0 to 64. The score increases as the emotion dysregulation increases.
- Depressive and anxiety symptoms were evaluated using the French version of the Hospital Anxiety and Depression Scale (HADS) [44,45]. The HADS is a 14-item self-rating scale that assesses the severity of depression (7 items) and anxiety (7 items). Each item is score between 0 (no, not at all) to 3 (yes, definitely). In this study, we used the total HAD depression subscore (ranging from 0 to 21) and the total HAD anxiety subscore (ranging from 0 to 21) as measures of depression and anxiety symptoms. The higher the score the higher the depression and anxiety.”
In addition, we have indicated the range of each score in Table 1.
Comment#6. In general, a more robust rationale for the study's approach and design needs to be provided with a clear and well-supported description and discussion of the findings between the quantitative and qualitative and how they relate to each other.
Reply:
We hope that the replies to the previous comments have added sufficient clarity to respond to this comment.
Comment#7. Minor point(s): please review the manuscript to ensure consistency in formatting. For example, when quoting the participants, italics were used with the exception of section 3.1.3.
Reply: We thank the reviewer for this comment. We have updated to manuscript so that all quotations from patients are in italics.

Reviewer 3 Report
Include in material and methods in the qualitative study section the categories of analysis that were explored
Author Response
Itemized list of reviewers comments with corresponding revisions or responses
Manuscript ID: ijerph-2067518
Decision: Major revision
Reviewer#3 Comments
Comment#1. Include in material and methods in the qualitative study section the categories of analysis that were explore
Reply: We have updated Section 2.5 lines 230-235 to include the following:
“The verbatims were analyzed with respect to various categories. Those based on the phenomenological approach explored the patient’s feelings (including categories around emotions such as sadness or fear). While those based on social interactions explored their point of view on organizational aspects during the lockdown (including categories about family and professional organizations, their adaptations to care access, and their health constraints).”
We would like to thank the editor and the reviewers for taking the time to review and comment our manuscript. We hope that the modifications proposed improved the manuscript’s clarity and quality.
Round 2
Reviewer 2 Report
I appreciate and thank the reviewers for their responsiveness to the issues raised.